# Mapping serotonergic dynamics using drug-modulated molecular connectivity in rats

Tudor M Ionescu[1], Mario Amend[1], Rakibul Hafiz[2], Andreas Maurer[1], Bharat Biswal[2], Hans F Wehrl[1]*, Kristina Herfert[1]*

[1]Werner Siemens Imaging Center, Department of Preclinical Imaging and Radiopharmacy, Eberhard Karls University Tuebingen, Tuebingen, Germany; [2]Department of Biomedical Engineering, New Jersey Institute of Technology, Newark, United States

## eLife Assessment

This **important** paper on measuring molecular connectivity using combined serotonin PET and resting-state fMRI provides both novel methods for studying the brain as well as insights into the effects of ecstasy administration. The methods are **convincing**, with the high anaesthetic dose used likely limiting network activity.

**\*For correspondence:**
wehrl@gmx.de (HFW);
kristina.herfert@med.uni-tuebingen.de (KH)

**Abstract** Understanding the complex workings of the brain is one of the most significant challenges in neuroscience, providing insights into normal brain function, neurological diseases, and the effects of potential therapeutics. A major challenge in this field lies in the limitations of traditional brain imaging techniques, which often capture only fragments of the complex puzzle of brain function. Our research employs a novel approach termed 'molecular connectivity' (MC), which combines the strengths of various imaging methods to provide a comprehensive view of how specific molecules, such as the serotonin transporter, interact across different brain regions and influence brain function. This innovative technique bridges the gap between functional magnetic resonance imaging (fMRI), known for its ability to monitor brain activity by tracking blood flow, and positron emission tomography (PET), which visualizes specific molecular changes. By integrating these methods, we can better understand how drugs influence brain function. Our study focuses on the application of dynamic [¹¹C]DASB PET scans to map the distribution of serotonin transporters, key players in regulating mood and emotions, and examines how these transporters are altered following exposure to methylenedioxymethamphetamine (MDMA), which is commonly known as ecstasy. Through a detailed comparison of MC with traditional measures of brain connectivity, we reveal significant patterns that closely align with physiological changes. Our results revealed clear changes in molecular connectivity after a single dose of MDMA, establishing a direct link between the effects of drugs on serotonin transporter occupancy and changes in the functional brain network. This work offers a novel methodology for the in-depth study of brain function at the molecular level and opens new pathways for understanding how drugs modulate brain activity.

## Introduction

The field of neuroscience has experienced remarkable advancements over the past decade, specifically propelled by the advent of innovative imaging techniques. The simultaneous application of PET and magnetic resonance imaging (MRI) has revolutionized our ability to assess brain function across

various physiological dimensions (*Wehrl et al., 2013*; *Judenhofer et al., 2008*). This is especially true in the context of drug effect evaluations. Combining PET with fMRI has opened many investigative avenues. Techniques such as resting-state functional connectivity (rs-FC) (*Biswal et al., 1995*) or pharmacological MRI (phMRI) (*Jonckers et al., 2015*) are now complemented by the ability of PET to quantify molecular changes, including alterations in receptor and transporter availability (*Sander et al., 2013*).

While fMRI provides high spatial and temporal resolution, the interpretation of its readout necessitates caution. The blood-oxygen-level-dependent (BOLD) signal in fMRI indirectly captures neuronal changes through neurovascular coupling (*Buxton, 2012*), revealing only the hemodynamic consequences of molecular-level drug effects. The integration of simultaneous PET acquisition can bridge this interpretative gap by offering essential molecular insights, particularly regarding transporter or receptor alterations. Typically, PET, used independently or simultaneously with fMRI, has been used mainly in pharmacological studies to illustrate quantitative shifts in neuroreceptor or transporter availability (*Silberbauer et al., 2019*). Several studies have also explored the interregional coherence of PET tracer signals (*Amend et al., 2019*; *Lanzenberger et al., 2012*; *Vanicek et al., 2017*), an approach akin to fMRI-derived rs-FC. While subject-level metabolic connectivity using [$^{18}$F]FDG-PET has been established through the temporal correlation of regional PET signals (*Wehrl et al., 2013*; *Amend et al., 2019*), studies employing transporter or receptor tracers have focused predominantly on interregional binding coherence across subjects using static scans (*Lanzenberger et al., 2012*; *Vanicek et al., 2017*). The concept and feasibility of molecular connectivity (MC) (*Hahn et al., 2019*; *Sala et al., 2023*) through the temporal correlation of the dynamic binding potentials of transporter or receptor tracers has yet to be explored.

In our study, we investigated the feasibility of deriving MC from dynamic [$^{11}$C]DASB-PET scans acquired simultaneously with fMRI in rats. We divided the rats into two cohorts, a baseline group and a pharmacological application group, which were exposed to 3,4-methylenedeoxymetamphetamine (MDMA). The baseline cohort assessed the feasibility of the novel methodology, contrasting it with traditional fMRI-derived rs-FC with a specific focus on temporal stability. We postulated that dynamic [$^{11}$C]DASB PET temporal fluctuations could be harnessed for connectivity data like BOLD signal can for hemodynamic rs-FC using seed-based and independent component analysis (ICA).

The second cohort, subjected to an MDMA challenge, allowed us to evaluate the utility of our novel approach. We aimed to outline the effects of MDMA by integrating the innovative MC concept with established analysis techniques. Our primary objective in this research was to elucidate the potential of PET-derived MC in conjunction with simultaneous PET/fMRI, exploring the avenues this methodology could open for future diagnostic and drug development studies.

## Results

### Comparability of MC and FC in spatial contexts and over time

We first aimed to evaluate whether MC align spatially with FC, possess similar graph theory properties, and provide consistent temporal readouts throughout the scan duration in the baseline group (*Figure 1*).

We found a moderate but significant correlation between the edge-level MC and FC (*r*=0.51, p<0.001; *Figure 1A and B*). Furthermore, both connectomes revealed small-world properties at the group level, with coefficients higher than 1 (*Figure 1C*). At the subject level, three molecular and four functional connectomes fell below the threshold of 1 for the small-world coefficient. Significant consistency was observed between early and late scan-derived connectomes (*Figure 1D* for FC, G for MC), with FC having a slight edge (*Figure 1E*, *r*=0.96) over MC (*Figure 1H*, *r*=0.8). While both FC and MC maintained steady correlation intensities, there was a negligible decline over the scan duration (*Figure 1F and I*).

### Deciphering the spatial characteristics of FC and MC using ICA

After establishing the feasibility of obtaining temporally stable readouts using the ROI-to-ROI approach, we employed a data-driven approach using ICA to compare the spatial characteristics of FC and MC (*Figure 2*).

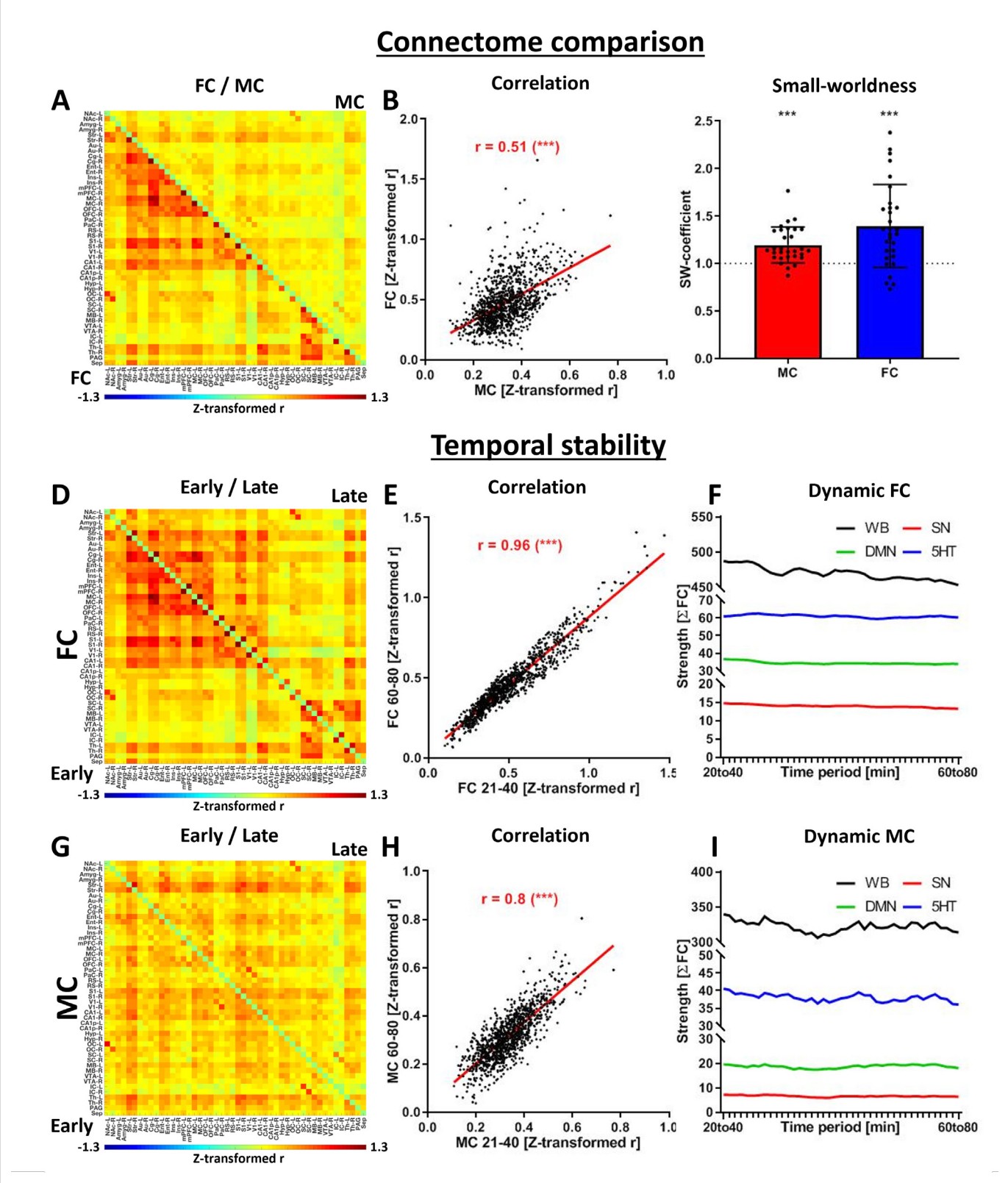

**Figure 1.** Evaluation of the seed-based molecular connectivity (MC). (**A**) Correlation matrix indicating whole-brain functional connectivity (FC) (beneath the diagonal) and MC (above the diagonal). Correlations not significant with multiple comparison corrections were set to zero (p<0.05, FWE correction). (**B**) Scatter plot and correlation between MC and FC edges. (**C**) Small-world coefficients for all subjects and group-level one-sample t-test against the value of 1 (SW >1 indicates small-world properties, data provided as mean ± SD, one-sample t-test to 1, *** p < 0.001). Comparison of (**D**) FC

*Figure 1 continued on next page*

*Figure 1 continued*

and (**G**) MC early (20–40 min after the start of the scan, below the diagonal) and late (60–80 min after the start of the scan, above the diagonal). The similarities of early and late readouts were quantified for both (**E**) FC and (**H**) MC. The temporal stability of both (**F**) FC and (**I**) MC was evaluated using a sliding window approach, including 20 min windows between 20 and 80 min after the start of the scan. Abbreviations: FC = fMRI-derived hemodynamic functional connectivity, MC = [$^{11}$C]DASB PET-derived molecular connectivity.

The online version of this article includes the following figure supplement(s) for figure 1:

**Figure supplement 1.** Detrending procedure of DVR-1 time courses.

We extracted ten group ICs from the fMRI data (*Figure 2A*), revealing known canonical resting-state networks, such as the posterior default-mode-like network (IC1-5, red), sensorimotor networks (IC1-5, green and purple), the anterior default-mode-like network (IC6-10, yellow) and the visual network (IC6-10, red). Given the unpredictability of the number of ICs suitable for MC IC extraction, we started with two components (*Figure 2B*). IC1 (orange) comprised both subcortical and cortical anterior brain regions, including the nucleus accumbens, amygdala, cingulate cortex, caudate putamen, orbitofrontal cortex and medial prefrontal cortex, whereas IC2 (blue) primarily received contributions from deeper posterior areas, such as the midbrain, thalamus, hypothalamus, periaqueductal gray cortex and medulla. Interestingly, when we extracted 10 independent components to mimic the number of components used for the FC data, the initial anterior component split into four different ICs, and the initial posterior IC split into three different ICs. Relatively clear spatial segregation can be observed for the newly formed ICs, for instance, the three posterior components extracted from specific regions (green, mainly from the medulla; red, from the hypothalamus; and part of the midbrain, blue, from the midbrain and thalamus).

## MDMA-induced changes in ICA-derived molecular connectivity

Next, we aimed to explore the relationship between molecular changes in SERT availability and the molecular connectome derived from the ICA induced by acute MDMA administration (*Figure 3*).

Two ICs extracted from the MC showed good overlap with regions associated with the SERT salience network (IC1) and the SERT subcortical network, comprising regions with intrinsically high SERT densities (IC2). Therefore, we defined the regions contributing strongly to IC2 as the SERT salience network (CPu, Cg, NAc, Amyg, Ins, mPFC) and those with strong signals in IC1 as the SERT subcortical network (VTA, Th, MB, PAG, Hypo). Interestingly, MDMA induced immediate strong decreases in all SERT subcortical network regions, and salience areas exhibited a delay of approximately 10 min (*Figure 3A*). Voxel-level analysis revealed clear spatial overlaps between early MDMA-responsive regions and those from the posterior IC, with delayed regions mirroring the anterior IC reminiscent of the SERT salience network (*Figure 3B*). To quantify the striking spatial similarity between the baseline independent components and the spatiotemporal characteristics of [$^{11}$C]DASB changes after MDMA exposure, we found highly significant correlations between the z scores of the posterior and anterior ICs and the t scores of late and early MDMA effects on [$^{11}$C]DASB alterations (p<0.001, *Figure 3C*).

Finally, we investigated the relationships between SERT availability changes and MC reductions following acute MDMA challenge (*Figure 4*).

We found that the decreases in MC and [$^{11}$C]DASB BP$_{ND}$ following MDMA application (*Figure 4A*) showed only a low correlation (*r*=0.29, *Figure 4B*). While decreases in SERT availability exhibited a strong anterior-posterior gradient, which was most pronounced in areas with high SERT availability, such as the MB, VTA, Pons, or PAG, the MC encompassed regions across the brain to similar extents. Specifically, while the significance of [$^{11}$C]DASB reductions in Ins was very low across the investigated regions, Ins presented the greatest effects among regions for MC. In contrast, strong SERT occupancy effects in IC and SC did not translate into very prominent reductions in the respective global MC of the two regions.

## MDMA-induced changes in seed-based molecular connectivity

Next, we performed a seed-based analysis to compare changes in FC and MC after an MDMA pharmacological challenge on the salience network and regions with high SERT binding (*Figure 5*).

We observed a decrease in SN FC (16% at the end of the scan, p=0.02, did not survive FDR correction) and almost constant FC of the subcortical network (<5% decrease, p=0.79) following MDMA, as shown in *Figure 5A* at the edge and node levels and *Figure 5C* at the network level. For MC, we

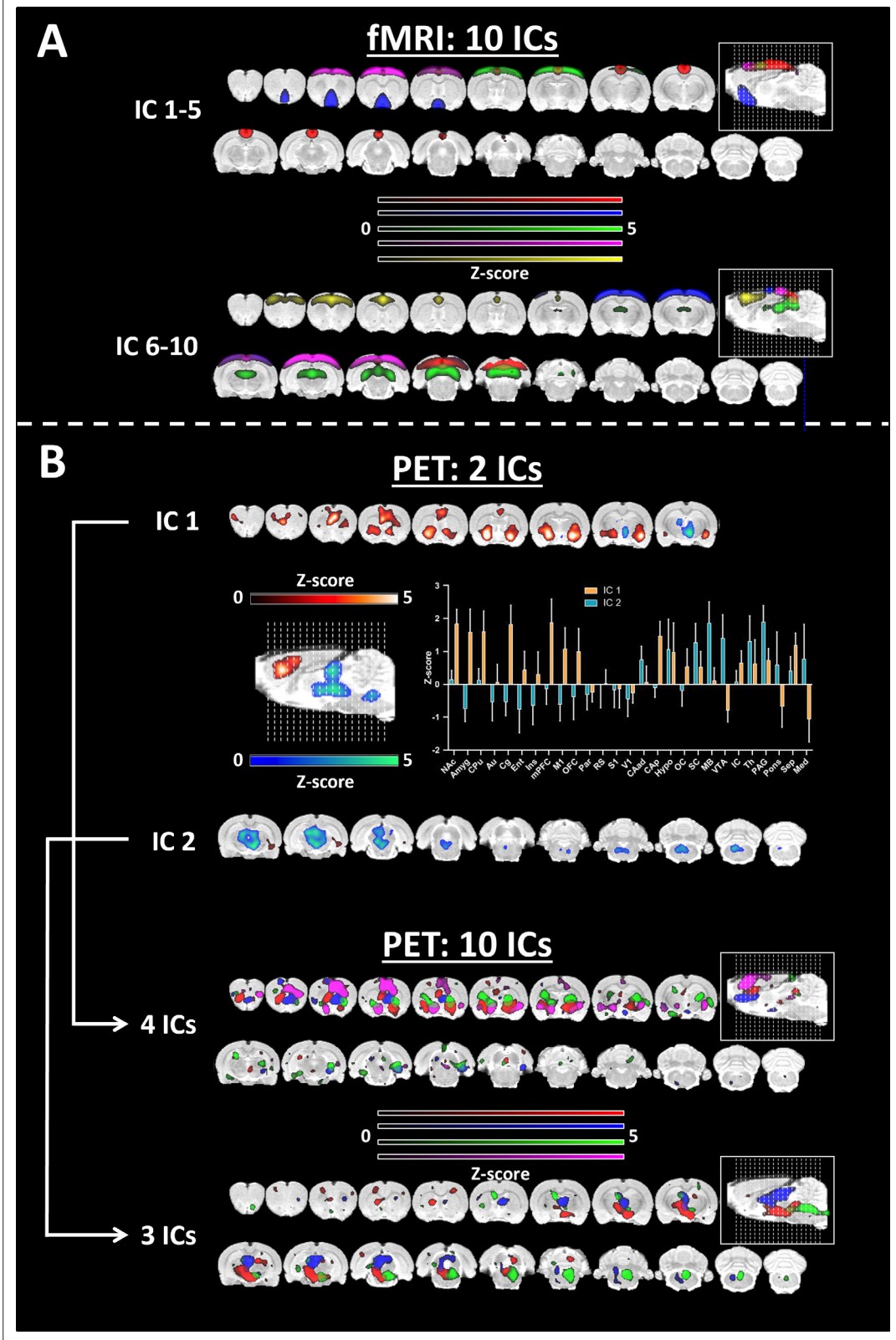

**Figure 2.** Group independent component analysis for functional connectivity (FC) and molecular connectivity (MC). (**A**) Independent component analysis (ICA) was performed over 10 components for functional magnetic resonance imaging (fMRI). (**B**) ICA performed over two components for [11C]DASB PET and regional quantification of the two derived components (mean ± SD over voxels). The ICA was repeated over 10 components. Four and

*Figure 2 continued*

three components showed good overlap with the two components defined above. All the components were thresholded at z>1.96 (p≤0.05). Abbreviations: FC = fMRI-derived hemodynamic functional connectivity, MC = [¹¹C] DASB PET-derived molecular connectivity.

observed profound reductions in the subcortical network (*Figure 5C and D*, p=0.009), emphasizing the acute and spatially specific effect of MDMA on MC.

## Discussion

The mammalian brain operates on diverse physiological, spatial, and temporal scales. FC via BOLD-fMRI offers insights into coherent functional brain networks, but its complexity and indirect link to neural activity highlight the need for more direct methodologies. In this context, the concept of MC using [¹¹C]DASB PET, as introduced in our study, provides a more direct and complementary perspective on brain organization and its response to external stimuli, such as MDMA.

### Physiological basis

Our findings suggest that [¹¹C]DASB binding reflects the interplay of serotonin levels and SERT dynamics (*Paterson et al., 2010*). In support of the competition model, evidence indicates that endogenous serotonin competes with tracers for binding sites, affecting tracer binding (*Yamamoto et al., 2007*). However, contrasting results from various studies highlight the complexity of this interaction (*Lundquist et al., 2007*; *Talbot et al., 2005*; *Lefevre et al., 2017*). The internalization model suggests that serotonin levels influence SERT internalization, impacting [¹¹C]DASB binding (*Paterson et al., 2010*; *Ramamoorthy and Blakely, 1999*; *Blakely et al., 1998*). While supporting evidence, further exploration is needed to fully understand these dynamics, especially during the resting state (*Raichle and Gusnard, 2002*). Some models on this topic have proposed a regulatory function of the raphé nuclei in maintaining serotonin fluctuations over several temporal scales at rest (*Salomon and Cowan, 2013*). Remarkably, fast microdialysis has resolved multiple spontaneous surges of up to 1500-fold of the basal serotonin occurring during 30 min intervals (*Yang et al., 2013*). Additionally, the same study indicated that SERT expression is essential for spontaneous surges and that reduced SERT drastically decreases serotonin spiking. Thus, it is feasible that the correlated temporal fluctuations captured by dynamic [¹¹C]DASB at least partly reflect the role of the serotonergic system in the resting activity of the brain. In addition, SERT regulation occurs over multiple time scales, ranging from milliseconds to hours, depending on the mechanism involved (*Lau and Schloss, 2012*). Rapid changes in SERT surface expression can be mediated by protein-protein interactions or posttranslational modifications (*Jayanthi et al., 2005*; *Steiner et al., 2008*), such as phosphorylation, which occur on a timescale of milliseconds to minutes. These processes dynamically modulate surface availability and function, allowing fine-tuned regulation of serotonin uptake even under resting-state conditions. Additionally, while slower processes involving endocytosis, recycling, and degradation typically occur over minutes to hours, subtle fluctuations in SERT trafficking and activity can still occur under basal conditions. These minor yet biologically relevant changes likely reflect ongoing homeostatic regulation essential for maintaining serotonergic balance. Therefore, tracer fluctuations observed during resting-state measurements should not be dismissed, as they may represent meaningful variations in SERT regulation that contribute to the fine control of serotonin clearance.

### Implications for the whole-brain serotonergic system

Our graph theory analysis revealed comparable whole-brain organization between the hemodynamic (BOLD-derived) and serotonergic (PET-derived) connectomes, although we found only a moderate correlation between the edges of the two measurements. Notably, the [¹¹C]DASB-based MC results showed a distinct binding pattern compared with BOLD-fMRI-derived FC. For FC, the ICA revealed classic RSNs, including the default-mode network, sensory network, and motor network, consistent with previous studies in rats (*Becerra et al., 2011*) and humans (*Fox et al., 2005*). In contrast, the PET data revealed two distinct anatomical components in the serotonergic system. One component included key subcortical areas, such as the brainstem, parts of the midbrain, and the thalamus, regions rich in SERT availability and central to serotonergic regulation. The other component comprised

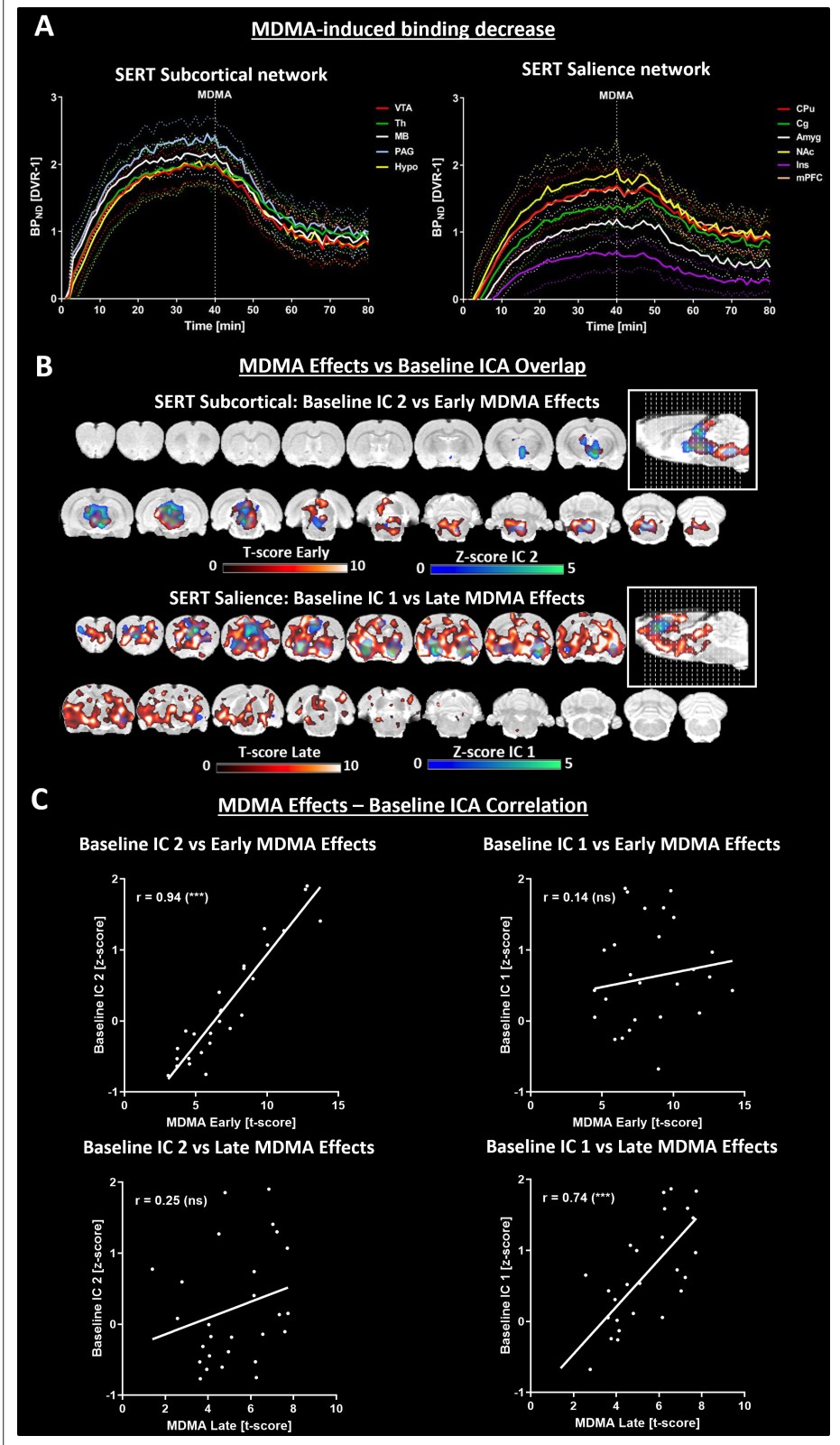

**Figure 3.** Comparison of methylenedioxymethamphetamine (MDMA)-induced [¹¹C]DASB alterations. (**A**) Left panel: Dynamic binding potentials of regions comprising the SERT subcortical network, defined by IC 1 in the validation cohort. Right panel: Dynamic binding potentials of regions comprising the SERT salience network, defined by IC 2 in the validation cohort (continuous lines indicate the means, and dotted lines indicate standard

*Figure 3 continued on next page*

*Figure 3 continued*

deviations). (**B**) Overlap between independent components extracted from the validation cohort (IC 1=SERT subcortical network, IC 2=SERT salience network) and the early and late effects of MDMA. (**C**) Pairwise correlations between regional z scores of the ICs extracted from the validation cohort and regional t scores of early and late MDMA effects. (*** indicates p<0.001, ns = not significant). Abbreviations: SERT = serotonin transporter, ICA = independent component analysis; for abbreviations of regions, please refer to the *Supplementary information*.

regions of the limbic system, such as the striatum, amygdala, insula, cingulate cortex, and prefrontal cortex, critical targets of serotonergic projections from the raphe nuclei. These components suggest a broader spatial reach of the serotonergic system than the typically described RSNs, reflecting its unique neurochemical architecture. In our analysis, PET ICs appeared less bilateral than fMRI ICs. This is likely due to the lower temporal resolution of PET (80 frames) than of fMRI (2400 frames), resulting in a reduced signal–to–noise ratio (SNR) and potentially affecting the stability and symmetry of the independent components.

Our results demonstrate that compared with FC, MDMA induces more pronounced changes in MC, particularly in regions associated with the SERT subcortical network. The distinct temporal dynamics of $BP_{ND}$ variations between these components may reflect the hierarchical organization of the serotonergic system. Specifically, the raphe nuclei, as the primary source of serotonin, are likely to exert more immediate modulation on posterior subcortical structures (IC2), whereas downstream effects on limbic and cortical regions (IC1) may occur more gradually. While these findings align with current neuroanatomical and molecular knowledge, the precise biological mechanisms driving these temporal differences remain unclear. Additional investigations are warranted to elucidate these mechanisms. Future studies combining direct measurements of serotonin levels with neuroimaging data will be critical to fully understanding these components' distinct roles and temporal profiles in regulating serotonergic function.

## Relevance to prior research

Previous studies evaluating the effects of MDMA on FC have indicated relatively few effects on FC (*Carhart-Harris et al., 2015*; *Roseman et al., 2014*). Our findings align with *Salvan et al., 2023*, who integrated molecular maps into fMRI data and demonstrated how individual serotonergic receptors contribute to network-level activity. Despite our differing methodologies, the receptor activity patterns found may also correspond to the independent components we found from the [$^{11}$C]DASB PET data. The dual regression approach reported by Salvan et al. was first reported in the context of pharmacology to delineate the receptor-specific effects of MDMA in humans (*Dipasquale et al., 2019*). The authors found that MDMA specifically decreased FC in the 5HT1A maps in areas that

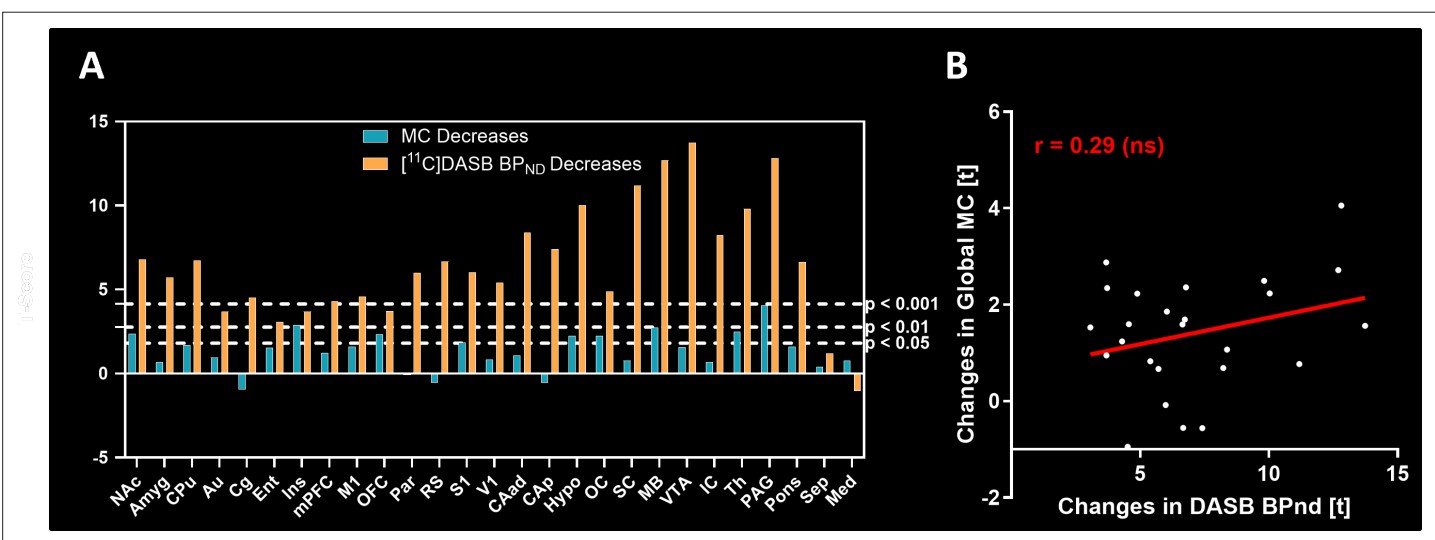

**Figure 4.** Comparison of molecular connectivity (MC) and $BP_{ND}$ changes following methylenedioxymethamphetamine (MDMA). (**A**) Reductions in $BP_{ND}$ following MDMA (orange) and MC strength (blue) were compared. (**B**) The correlation coefficient of the regional T scores was low (*r*=0.29).

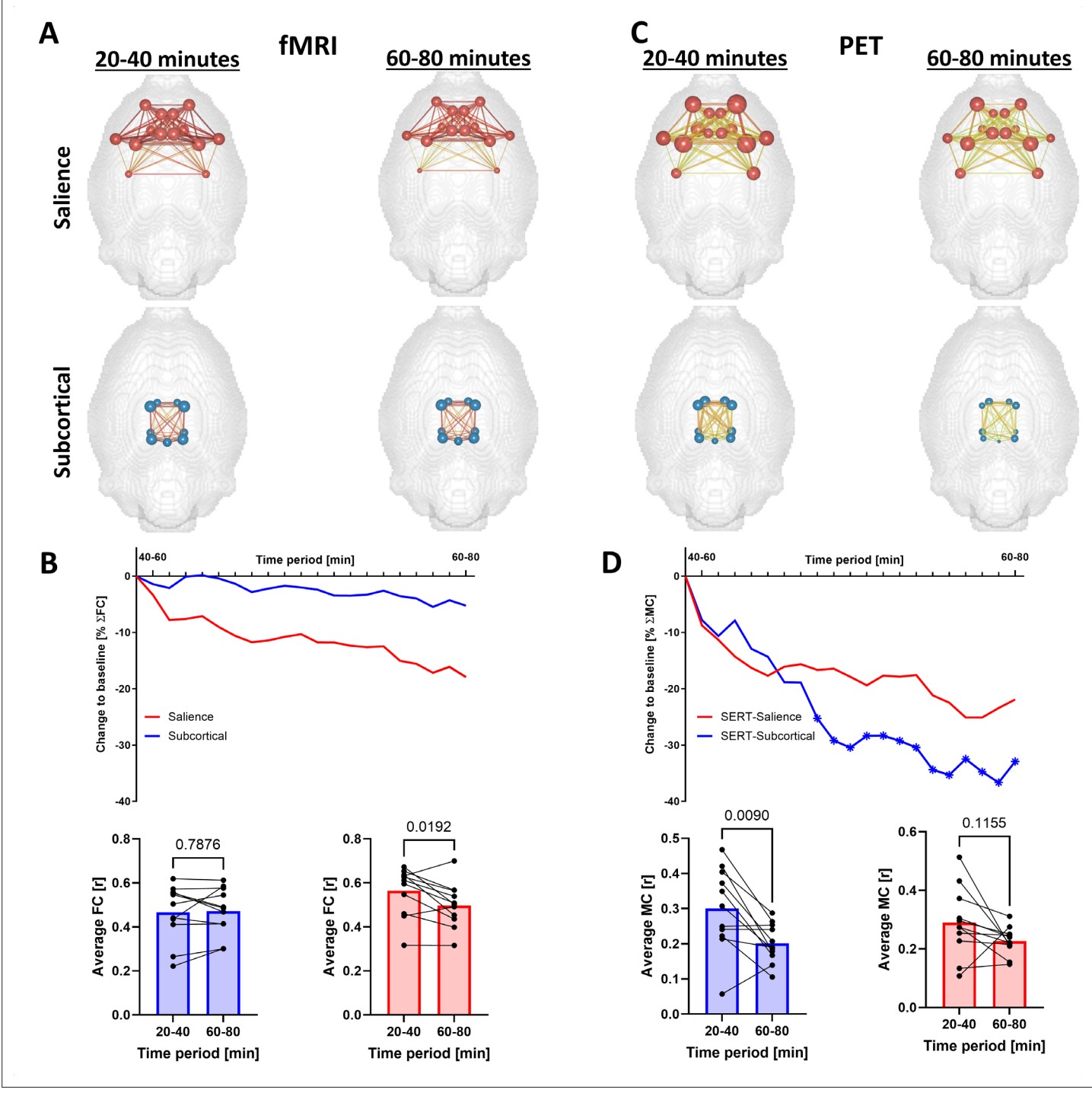

**Figure 5.** Methylenedioxymethamphetamine (MDMA) effects on seed-based functional connectivity (FC) and molecular connectivity (MC) of the salience and subcortical networks. (**A**) FC and (**C**) MC brain networks depicting the edge and node strengths of the salience network and subcortical network at baseline (20–40 min after the start of the scan) and after MDMA (60-80 min after the start of the scan). (**B**) FC and (**D**) MC time-resolved salience and subcortical network strengths computed by sliding windows. Individual values are provided in the bar graphs for the baseline and final post-MDMA periods (paired t-tests, numbers indicate p-values). Asterisks indicate significant (p<0.05, FDR-corrected) changes from baseline (time point zero, corresponding to 20–40 min after the start of the scan).

could be ascribed to the human salience network, such as the insula and a collection of medial cortical regions. While not reaching significance, we observed a trend towards reduced salience FC and MC in our data.

The increased activity in limbic and cortical structures when controlling for vascular effects being accompanied by decreased salience connectivity, indicates that, while neurons become more active through the drug, they do so in an incoherent manner, which would be in line with the hyperactive yet abnormal behavior reported for MDMA abuse. Importantly, the potent vascular effects that play a role in modulating the amplitude of the BOLD-fMRI signal may also influence FC readouts, although the extent of such effects is difficult to estimate (*Ionescu et al., 2023*).

Over the past decade, PET studies using [$^{11}$C]DASB have focused on the associations of serotonin transporter (SERT) availability across different brain regions, revealing altered interregional SERT connections in patient cohorts, posttreatment changes, and a predictive capacity for treatment response (*Lanzenberger et al., 2012*; *Vanicek et al., 2017*; *James et al., 2017*; *Hahn et al., 2014*; *Hahn et al., 2010*). Our study builds upon this foundation, revealing significant within-subject temporal associations in [$^{11}$C]DASB binding and demonstrates more pronounced alterations in MC than in FC induced by drugs, thus highlighting the enhanced utility of our method.

## Study limitations and future directions

One limitation of our study is that our experimental protocols predate the recently published consensus recommendations for rat fMRI (*Grandjean et al., 2023*), particularly concerning anesthesia and preprocessing pipelines. Using isoflurane anesthesia, although common at the time of data acquisition, introduces a potential confound due to its known neuronal activity effects. However, we previously demonstrated that isoflurane at 1.3% maintains stable physiological parameters and avoids burst suppression (*Ionescu et al., 2021b*), a concern at higher doses. Furthermore, other studies have reported that low-dose isoflurane remains feasible for resting-state functional connectivity studies (*Hutchison et al., 2014*). While isoflurane, a GABA-A agonist, could theoretically interact with the mechanisms of MDMA in the brain, we found no evidence in the literature suggesting significant cross-talk between these substances. Future studies employing medetomidine-based protocols may help minimize this potential confounding factor.

Additionally, the number of animals used for our pharmacological study with MDMA was low compared to the resting-state cohort. While future studies with higher group sizes may be valuable to confirm and expand our findings, the consistency of effects between subjects indicated in our data points towards the stability of our results even at the employed group size.

With respect to data preprocessing, we retained the same pipeline used in our prior publications (*Ionescu et al., 2023*; *Ionescu et al., 2021a*) to maintain methodological consistency. While we recognize the advantages of adopting standardized preprocessing, as outlined in the consensus guidelines, this approach ensures comparability with our previous analyses. To facilitate further investigation, we have made the full dataset publicly available (see Data availability statement), enabling reanalysis with updated pipelines or additional explorations of this dataset.

Nonetheless, we provide a strong rationale for considering such analyses in future studies. First, we show that at rest, the data are reliable, temporally stable, and exhibit similar graph theory metrics to traditionally calculated functional connectomes. Second, despite comparable network-level organizational properties, at rest, MC is only moderately correlated with FC, indicating the complementary nature of both readouts. Third, resting-state MC ICs correlate well with SERT occupancy changes induced by the MDMA challenge. Finally, we show that the changes that MDMA elicits on the MC are complementary to standardly calculated [$^{11}$C]DASB BP$_{ND}$ alterations. Our data indicate that while changes in BP$_{ND}$ are more pronounced in regions with higher baseline SERT availabilities, MC reveals a more globally distributed measure of tracking serotonergic changes since regions, such as the insula, with relatively low SERT expression, are strongly affected. However, our multimodal imaging approach, although powerful, cannot fully decipher the mechanisms of interregional coherence in PET time courses. Therefore, studies employing more direct sensitive methods to measure neurotransmitter release could provide deeper insights into the molecular processes underpinning our observations.

This study significantly contributes to integrating molecular data into connectomic frameworks, demonstrating that subject-level MC are reliable and complementary to FC in both resting and pharmacologically challenged states. Our research provides a strong foundation for future investigations

into the value and generalizability of PET-derived MC, particularly for understanding drug-induced brain-wide molecular network changes.

## Materials and methods

Our study reevaluates two datasets previously published by our group (*Ionescu et al., 2023*; *Ionescu et al., 2021a*) to explore FC and MC simultaneously at baseline (*Ionescu et al., 2021a*) and following MDMA administration (*Ionescu et al., 2023*). Please refer to these earlier publications for detailed descriptions of the animal handling methods, experimental setups, and data acquisition procedures.

### Animals

A total of 41 male Lewis rats were obtained from Charles River Laboratories (Sulzfeld, Germany). Thirty rats (354±37 g, corresponding to 11 wk of age) underwent baseline [$^{11}$C]DASB PET/fMRI scans without any pharmacological intervention, whereas 11 rats (365±19 g, corresponding to 11 wk of age) underwent [$^{11}$C]DASB PET/fMRI scans, including an acute MDMA challenge. All experiments were in compliance with German federal regulations for experimental animals and received approval from the Regierungspräsidium Tübingen.

### Simultaneous PET/fMRI experiments

The rats were subjected to simultaneous PET/fMRI experiments involving 1.3% isoflurane anesthesia, tail vein catheterization, positioning on a temperature-controlled bed, and monitoring of vital signs. The scans were performed using a 7T small-animal ClinScan MRI scanner (Bruker Biospin, Ettlingen, Germany) with a custom-developed PET insert (*Disselhorst et al., 2022*). For [$^{11}$C]DASB PET, we employed a bolus plus constant infusion protocol ($k_{bol}$ = 38.7 min) and reconstructed the scans in 1 min time frames. The scanning protocol and sequence parameters are outlined in detail in our previous publication (*Ionescu et al., 2023*). The MDMA cohort received a pharmacological MDMA challenge of 3.2 mg/kg intravenously 40 min after tracer injection.

### Data preprocessing and analysis

Data preprocessing followed established protocols, including steps such as realignment, mask creation, coregistration, spatial normalization, signal cleaning, and spatial smoothing, as detailed in our previous work (*Ionescu et al., 2023*). For the MDMA dataset, PET scans were analyzed for early and late effects post-challenge using the general linear model (GLM) available in SPM. For both datasets, the baseline was defined as 30–40 min after the start of the scan. For the fMRI data, a first-level analysis was applied to the individual scans without a high-pass filter (the filter was set to 'Inf'). Statistical parametric maps were generated after GLM parameter estimation using contrast vectors. The group-level analysis involved a one-sample t-test on the subject-level statistical parametric maps (p<0.05, one-sided, familywise error/FWE adjusted).

Static PET scans were generated by summing dynamic frames over defined periods for 10 min periods after the MDMA challenge (50–60 min to investigate early effects and 70–80 min to investigate late MDMA effects). Two-sample t-maps were calculated between the normalized [$^{11}$C]DASB uptakes of (1) the baseline scan period and the early effect period and (2) the early effect period and the late effect period (p<0.05, FWE-adjusted).

All group-level t-maps were subjected to voxelwise signal quantification to determine the regional contributions of 48 regions selected according to the Schiffer atlas (*Schiffer et al., 2006*). The average t-scores and standard deviations of all voxels were calculated.

### Functional connectivity analysis

FC was determined using a seed-based analysis approach. The mean time series of the preprocessed BOLD-fMRI signals for each dataset across all regions (refer to *Table 1* for the list of regions) were extracted using the SPM toolbox Marseille Boîte À Région d'Intérêt (MarsBaR). Pairwise Pearson's correlation coefficients were calculated between each pair of mean regional time series for every dataset. The Pearson's r coefficients were converted into z values using Fischer's transformation for group-level analysis. Fischer's z-transformed correlation coefficients were then used to generate mean correlation matrices for both cohorts (*Xia et al., 2013*).

**Table 1.** Brain regions included in the Schiffer rat brain atlas, including their respective volumes and abbreviations.

| Brain region (ROI) | Hemisphere | ROI volume [mm³] | Position on the correlation matrix | Abbreviation |
|---|---|---|---|---|
| Nucleus Accumbens | left | | 1 | NAc |
| | right | 7.944 | 2 | |
| Amygdala | left | | 3 | Amyg |
| | right | 21.120 | 4 | |
| Dorsal Striatum | left | | 5 | Str |
| | right | 43.552 | 6 | |
| Auditory Cortex | left | | 7 | Au |
| | right | 27.520 | 8 | |
| Cingulate Cortex | left | | 9 | Cg |
| | right | 14.480 | 10 | |
| Entorhinal Cortex | left | | 11 | Ent |
| | right | 59.016 | 12 | |
| Insular Cortex | left | | 13 | Ins |
| | right | 21.128 | 14 | |
| Medial Prefrontal Cortex | left | | 15 | mPFC |
| | right | 6.304 | 16 | |
| Motor Cortex | left | | 17 | M1 |
| | right | 32.608 | 18 | |
| Orbitofrontal Cortex | left | | 19 | OFC |
| | right | 18.936 | 20 | |
| Parietal Cortex | left | | 21 | PaC |
| | right | 7.632 | 22 | |
| Retrosplenial Cortex | left | | 23 | RS |
| | right | 18.920 | 24 | |
| Somatosensory Cortex | left | | 25 | S1 |
| | right | 71.600 | 26 | |
| Visual Cortex | left | | 27 | V1 |
| | right | 36.136 | 28 | |
| Anterodorsal Hippocampus | left | | 29 | CA1 |
| | right | 25.064 | 30 | |
| Posterior Hippocampus | left | | 31 | CA1-p |
| | right | 9.784 | 32 | |
| Hypothalamus | left | | 33 | Hyp |
| | right | 18.352 | 34 | |
| Olfactory Cortex | left | | 35 | OC |
| | right | 14.008 | 36 | |
| Superior Colliculus | left | 7.136 | 37 | SC |
| | right | | 38 | |

*Table 1 continued*

| Brain region (ROI) | Hemisphere | ROI volume [mm³] | Position on the correlation matrix | Abbreviation |
|---|---|---|---|---|
| | left | | 39 | MB |
| Midbrain | right | 11.448 | 40 | |
| | left | | 41 | VTA |
| Ventral Tegmental Area | right | 5.528 | 42 | |
| | left | | 43 | IC |
| Inferior Colliculus | right | 5.744 | 44 | |
| | left | | 45 | Th |
| Thalamus | right | 30.712 | 47 | |
| Periaqueductal Gray | - | 9.904 | 47 | PAG |
| Septum | - | 9.36 | 48 | Sep |

## Molecular connectivity analysis

The mean [¹¹C]DASB signal from the preprocessed PET datasets was extracted from the designed regions, including the 48 regions used for fMRI data analysis and the cerebellum using MarsBaR. Binding potentials were calculated framewise for all dynamic PET scans using the DVR-1 (*Equation 1*) to generate regional $BP_{ND}$ values with the cerebellar gray matter as a reference region, which our earlier studies demonstrated to be the most appropriate for this tracer in rats (*Walker et al., 2016*; *Walker et al., 2020*):

$$BP_{ND} = \frac{V_T - V_{ND}}{V_{ND}} = \frac{V_T}{V_{ND}} - 1 = DVR - 1 \tag{1}$$

where:

- $BP_{ND}$ is the binding potential
- $V_T$ is the total volume of distribution
- $V_{ND}$ is the volume of distribution in a reference tissue
- $DVR$ is the relative volume of distribution

To calculate the MC, we discarded the first 20 min of every scan, which were dominated by perfusion effects, and applied a detrending approach to the remaining 60 min to obtain temporally stable values (for further details, please refer to the Supplementary Methods and *Figure 1—figure supplement 1*). The $BP_{ND}$ time courses were then used to calculate the MC as described above for fMRI: ROI-to-ROI subject-level correlation matrices between all regional time courses were generated, and z-transformed correlation coefficients were used to calculate mean correlation matrices.

## Independent component analysis

Group ICA (GIFT toolbox, MIALAB, University of New Mexico, Albuquerque, NM, USA) was used for ICA in the baseline group. The fMRI and PET preprocessed datasets were investigated between 30 and 80 min after the start of data acquisition. For fMRI, we selected ten independent components, while we started with two components for PET and increased the number to ten components to thoroughly dissect the varying components within the signal. The components were thresholded at a z-value ≥1.96 (p-value ≤0.05) (*Di and Biswal, 2012*), and average z-scores and standard deviations were calculated for each component. The physiological significance of the components was further explored by contrasting them with the regional [¹¹C]DASB changes induced by MDMA. Accordingly, the z scores of the independent components generated in the baseline cohort were correlated with the early and late regional [¹¹C]DASB changes induced by MDMA, measured using t scores.

## Schiffer brain atlas

The Schiffer brain atlas was used to parcelate the brains for ROI seed-based analysis into 23 bilateral regions, the periaqueductal gray (PAG) and the septum (Sep), resulting in a total number of 48 brain regions. A list of all regions and their respective abbreviations is provided in **Error! Reference source not found**. *Table 1*.

## Radiotracer synthesis

[11C]DASB was synthesized similarly to the procedure reported by *Wilson et al., 2000*. Briefly, [11C]MeI was trapped in a solution of 2 mg precursor in 500 µl DMSO. After heating to 100 °C for 2 min, the reaction was diluted with 1.5 ml HPLC eluent (3 mM $Na_2HPO_4$ containing 64% MeCN) and purified on a Luna C18(2) column (250 mm × 10 mm, Phenomenex). The isolated peak was diluted with 70 ml water containing 20 mg sodium ascorbate, loaded onto a conditioned Strata-X cartridge (Phenomenex), eluted with 0.5 ml ethanol, and diluted with 5 ml phosphate-buffered saline.

## DVR-1 detrending

We applied a piecewise linear detrend to remove remaining tracer uptake dynamics from the dynamic DVR-1 time series and its effect on the artificial inflation of correlation values in the early parts of the scan, before reaching a steady state, while only discarding the first 20 min of every scan, which were too strongly dominated by tracer uptake effects (*Figure 1—figure supplement 1–*).

## Acknowledgements

This research was supported by funds from the Eberhard Karls University Tübingen Faculty of Medicine (fortüne 2209-0-0 and 2409-0-0) to HFW, from Bundesministerium für Bildung und Forschung (BMBF, Grant No. 01GQ1415) to HFW and from the Carl Zeiss Foundation to KH.

## Additional information

### Funding

| Funder | Grant reference number | Author |
|---|---|---|
| Eberhard Karls Universität Tübingen | fortüne 2209-0-0 | Hans F Wehrl |
| Eberhard Karls Universität Tübingen | fortüne 2409-0-0 | Hans F Wehrl |
| Bundesministerium für Bildung und Forschung | 01GQ1415 | Hans F Wehrl |
| Carl-Zeiss-Stiftung | | Kristina Herfert |

The funders had no role in study design, data collection and interpretation, or the decision to submit the work for publication.

### Author contributions

Tudor M Ionescu, Conceptualization, Software, Formal analysis, Validation, Visualization, Methodology, Writing – original draft, Writing – review and editing; Mario Amend, Supervision, Investigation, Writing – review and editing; Rakibul Hafiz, Software, Methodology, Writing – review and editing; Andreas Maurer, Methodology, Writing – review and editing; Bharat Biswal, Conceptualization, Software, Supervision, Funding acquisition, Methodology, Writing – review and editing; Hans F Wehrl, Conceptualization, Supervision, Funding acquisition, Investigation, Project administration, Writing – review and editing; Kristina Herfert, Resources, Supervision, Funding acquisition, Project administration, Writing – review and editing

### Author ORCIDs

Tudor M Ionescu ⬤ https://orcid.org/0000-0002-7623-9566
Hans F Wehrl ⬤ https://orcid.org/0000-0003-4692-3209

Kristina Herfert [ID] https://orcid.org/0000-0003-0231-7717

### Ethics

All experiments were in compliance with German federal regulations for experimental animals and received approval from theRegierungspräsidium Tübingen.

Reviewer #1 (Public review): https://doi.org/10.7554/eLife.97864.3.sa1
Reviewer #2 (Public review): https://doi.org/10.7554/eLife.97864.3.sa2
Author response https://doi.org/10.7554/eLife.97864.3.sa3

---

## Additional files

### Supplementary files

MDAR checklist

### Data availability

The data are openly available at Dryad (https://doi.org/10.5061/dryad.6djh9w1bf).

The following dataset was generated:

| Author(s) | Year | Dataset title | Dataset URL | Database and Identifier |
|---|---|---|---|---|
| Ionescu T, Amend M, Hafiz R, Maurer A, Biswal B, Wehrl HF, Herfert K | 2024 | Data from: Mapping serotonergic dynamics using drug-modulated molecular connectivity | https://doi.org/10.5061/dryad.6djh9w1bf | Dryad Digital Repository, 10.5061/dryad.6djh9w1bf |

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
