## [Editor Report · eLife Assessment]

This **important** paper on measuring molecular connectivity using combined serotonin PET and resting-state fMRI provides both novel methods for studying the brain as well as insights into the effects of ecstasy administration. The methods are **convincing**, with the high anaesthetic dose used likely limiting network activity.

---

## [Referee Report · Reviewer #1 (Public review)]

This paper by Ionescu et al. applies novel brain connectivity measures based on fMRI and serotonin PET both at baseline and following ecstasy use in rats. There are multiple strengths to this manuscript. First, the use of connectivity measures using temporal correlations of 11C-DASB PET, especially when combined with resting state fMRI, is highly novel and powerful. The effects of ecstasy on molecular connectivity of the serotonin network and salience network are also quite intriguing.

The authors discussed their use of high-dose (1.3%) isolfurane in the context of a recent consensus paper on rat fMRI (Grandjean et al., "A Consensus Protocol for Functional Connectivity Analysis in the Rat Brain.") which found that medetomidine combined with low dose isoflurane provided optimal control of physiology and fMRI signal. The authors acknowledge their suboptimal anaesthetic regimen, which was chosen before the publication of the consensus paper. This likely explains, in part, why fMRI ICs in figure 2A appear fairly restricted.

The PET ICs appear less bilateral than the fMRI ICs, which the authors attribute to lower SNR.

---

## [Referee Report · Reviewer #2 (Public review)]

Summary:

The article aims to describe a novel methodology for the study of brain organization, in comparison to fMRI functional connectivity, under rest vs. controlled pharmacological stimulation.

Strengths:

Solid study design with pharmacological stimulation applied to assess the biological significance of functional and (novel) molecular connectivity estimates.

Provides relevant information on the multivariate organization of serotoninergic system in the brain.

Provides relevant information on the sensitivity of traditional (univariate PET analysis, fMRI functional connectivity) and novel (molecular connectivity) methods in measuring pharmacological effects on brain function.

Comments on revisions:

I thank the authors for carefully addressing my comments and in particular for the interesting insights added to the discussion.

I have just one last remark pertaining to the point of the sample size: rats undergoing the MDMA acute challenge constitute a relatively small sample (N=11); I feel there is a certain risk the results presented might not be particularly replicable. Could the authors prove the stability of their (main) results by randomly iterating the individuals included in their sample (e.g. via permutation tests)? Alternatively, including at least a justification of the sample size in the context of the available evidence would be valuable.

---

## [Author Response]

The following is the authors’ response to the original reviews.

**Reviewer 1:**
Comment 1- I would like the authors to discuss and justify their use of high-dose (1.3%) isolfurane. A recent consensus paper on rat fMRI (Grandjean et al., "A Consensus Protocol for Functional Connectivity Analysis in the Rat Brain.") found that medetomidine combined with low dose isoflurane provided optimal control of physiology and fMRI signal. To overcome any doubts about the effects of the high-dose anaesthetic I'd encourage the authors to show the results of their functional connectivity specificity using the same or similar image processing protocol as described in that consensus paper. This is especially true since the fMRI ICs in Figure 2A appear fairly restricted.

We thank the reviewer for their insightful comments. We agree that the combination of medetomidine and isoflurane, as recommended by Grandjean et al. in their consensus paper, provides superior physiological stability and fMRI signal quality, and should indeed be considered the preferred protocol for future studies. In fact, we have adopted this combination in our subsequent research [1]. However, the data acquired in the present study were acquired prior to the publication of the consensus recommendations and have been previously published [2, 3]. While isoflurane is not the ideal anesthetic for functional connectivity studies, we have demonstrated in earlier work [4], that using isoflurane at 1.3% maintains stable physiological parameters and avoids burst suppression, a key issue with higher isoflurane doses.

Regarding preprocessing, we acknowledge the importance of standardized approaches as outlined in the consensus paper. However, to maintain methodological consistency with our prior work, we retained the original preprocessing pipeline for this study. This decision ensures comparability with our previous analyses. To address the reviewer’s concerns and encourage further verification, we have uploaded the full dataset to a public repository (as suggested in Comment 4). This will enable other researchers to reanalyze the data using updated preprocessing pipelines or explore additional analyses.

We have updated the manuscript discussion (page 19) to clearly acknowledge these points:

“One limitation of our study is that our experimental protocols predate the recently published consensus recommendations for rat fMRI [42], particularly concerning anesthesia and preprocessing pipelines. The use of isoflurane anesthesia, although common at the time of data acquisition, introduces a potential confound due to its known effects on neuronal activity. However, we previously demonstrated that isoflurane at 1.3% maintains stable physiological parameters and avoids burst suppression [43], a concern at higher doses. Furthermore, other studies have reported that low-dose isoflurane remains feasible for resting-state functional connectivity studies [44]. While isoflurane, as a GABA-A agonist, could theoretically interact with the mechanisms of MDMA in the brain, we found no evidence in the literature suggesting significant cross-talk between these substances. Future studies employing medetomidine-based protocols may help minimize this potential confound.

Regarding data preprocessing, we chose to retain the same pipeline used in our prior publications [13, 14] to maintain methodological consistency. While we recognize the advantages of adopting standardized preprocessing as outlined in the consensus guidelines, this approach ensures comparability with our previous analyses. To facilitate further investigation, we have made the full dataset publicly available (see Data Availability Statement), enabling reanalysis with updated pipelines or additional explorations of this dataset.”

Comment 2 - I'd also be interested to read more about why the cerebellum was chosen as a reference region, given that serotonin is highly expressed in the cerebellum, and what effects the choice of reference region has on their quantification.

This is something we ourselves have examined in a paper, dedicated to determine the most suitable reference region for [11C]DASB, and while the reviewer is correct in saying there is also serotonin in the cerebellum, we found the lowest binding for this tracer in the cerebellar gray matter, recommending this region as a valid reference area. (“Displaceable binding of (11)C-DASB was found in all brain regions of both rats and mice, with the highest binding being in the thalamus and the lowest in the cerebellum. In rats, displaceable binding was largely reduced in the cerebellar cortex”, please refer to [5]).

We amended our materials and methods part to specify that we had shown in this previous publication that the cerebellar gray matter is appropriate as a reference region (page 6):

“Binding potentials were calculated frame-wise for all dynamic PET scans using the DVR-1 (equation 1) to generate regional BP_ND_ values with the cerebellar gray matter as a reference region, which our earlier studies have demonstrated to be the most appropriate for this tracer in rats [5, 6]:”

Comment 3 - The PET ICs appear less bilateral than the fMRI ICs. Is that simply a thresholding artefact or is it a real signal?

We thank the reviewer for this observation. The reduced bilaterality of PET ICs compared to fMRI ICs is likely due to the inherent limitation in the temporal resolution of PET, which provides significantly fewer frames (100 frames compared to 3000 frames for fMRI). This lower temporal resolution leads to reduced signal-to-noise ratio when computing the ICA, which can affect the stability and symmetry of the ICs during ICA computation, particularly at higher IC numbers. While thresholding may also a minor role, we believe the primary factor is poorer SNR associated with the PET data. We have clarified this point in the discussion section (page 17) as follows:

“In our analysis, PET ICs appeared less bilateral than fMRI ICs. This is likely due to the lower temporal resolution of PET (100 frames) compared to fMRI (3000 frames), resulting in reduced signal-to-noise ratio (SNR) and potentially affecting the stability and symmetry of the independent components.”

Comment 4 - "The data will be made available upon reasonable request" is not sufficient - please deposit the data in an open repository and link to its location.

We agree with the request of the reviewer and uploaded the data to a Dryad repository. We amended our Data Availability Statement accordingly.

Comment 5 (recommendation) - Please add the age and sex of the rats in lines 92-97.

Amended.

Comment 6 (recommendation) - There are multiple typos throughout the manuscript - for example, "z-vlaue" on line 164, "negligable" on line 194, etc.. Sometimes the 11 in 11C is superscripted, sometimes it isn't. This paper would benefit from a careful proofread.

Thank you for pointing this out. We sent the manuscript for language and grammar editing to AJE (see certificate).

**Reviewer 2:**
Comment 1 - While the study protocol is referenced in the paper, it would be useful to at least report whether the study uses bolus, constant infusion, or a combination of the two and the duration of the frames chosen for reconstruction. Minimal details on anesthesia should also be reported, clarifying whether an interaction between the pharmacological agent for anesthesia and MDMA can be expected (whole-brain or in specific regions).

We fully agree that this would improve the readability of our manuscript and added the information to the materials and methods and discussion accordingly. Please refer to page 4/5.

Comment 2 - Some terminology is used in a bit unclear way. E.g. "seed-based" usually refers to seed-to-voxel and not ROI-to-ROI analysis, or e.g. it is a bit confusing to have IC1 called SERT network when in fact all ICs derived from DASB data are SERT networks. Perhaps a different wording could be used (IC1 = SERT xxxxx network; IC2 = SERT salience network).

Based on the reviewer´s suggestion, we suggest to rename IC1 and IC2 according to their anatomical and functional characteristics (page 13):

“IC1 = SERT Salience Network: This name highlights the involvement of the regions typically associated with the salience network (e.g., CPu, Cg, NAc, Amyg, Ins, mPFC), which play key roles in emotional and cognitive processing.”

“IC2 = SERT Subcortical Network: This name reflects the involvement of subcortical regions which play a role in arousal, stress response, and autonomic regulation, which are heavily modulated by serotonin in areas like the hypothalamus, PAG, and thalamus.”

Comment 3 - The limited sample size for the rats undergoing pharmacological stimulation which might make the study (potentially) not particularly powerful. This could not be a problem if the MDMA effect observed is particularly consistent across rats. Information on inter-individual variability of FC, MC, and BP_ND_ could be provided in this regard.

We thank the reviewer for raising this point. To address the concern about limited sample size and inter-individual variability, we have added this information to Figures 5 B and D. Regarding the BP_ND_ variability, the dotted lines in Figure 3 indicate the standard deviation in the regional BP_ND_, however, this was not clearly stated in the original figure description. We have now amended the figure legend to explicitly clarify this point.

Comment 4 (recommendation) - "Our research employs a novel approach named "molecular connectivity" (MC), which merges the strengths of various imaging methods to offer a comprehensive view of how molecules interact within the brain and affect its function." I'd recommend rephrasing to "..how molecular interact across different areas within the brain..". Molecular connectivity is a potentially ambiguous term (used to study interactions across different molecules (in the same compartment/environment) vs. to study interactions across the same molecules in different areas). I'd add a couple of references to help the reader disambiguate too (e.g. https://pubmed.ncbi.nlm.nih.gov/30544240/ , https://pubmed.ncbi.nlm.nih.gov/36621368/)

We appreciate the reviewer’s suggestion and agree that the term "Molecular Connectivity" could be ambiguous. To clarify, we rephrased the description to emphasize that our approach specifically examines interactions of the same molecule (i.e., serotonin transporter) across different brain regions, rather than interactions between different molecules within the same environment. We propose the following revised text (page 2):

“Our research employs a novel approach termed molecular connectivity (MC), which combines the strengths of various imaging methods to provide a comprehensive view of how specific molecules, such as the serotonin transporter, interact across different brain regions and influence brain function.”

Additionally, we will incorporate the suggested references to help the reader further contextualize the use of this term.

Comment 5 - In the methods, it is not clear if for MC the authors also compute ROI-to-ROI correlations or only ICA.

Thank you for highlighting this point. To clarify, our MC analysis, includes both ROI-to-ROI correlations and ICA. Specifically, as described at the end of the “Molecular Connectivity Analysis” subchapter, we compute ROI-to-ROI correlations using the following steps: 1. The first 20 minutes of each scan are discarded to account for perfusion effects. 2. A detrending approach is applied to the remaining 60 minutes of BP_ND_ time courses. 3. ROI-to-ROI calculations are then calculated and organized into subject-level correlation matrices, which are subsequently z-transformed to generate mean correlation matrices across subjects.

We revised the methods section to explicitly state that both ROI-to-ROI correlations and ICA are integral components of the MC analysis to ensure this point is clear to readers (page 6).

“The BP_ND_ time courses were then used to calculate MC as described above for fMRI: ROI-to-ROI subject-level correlation matrices between all regional time courses were generated and z-transformed correlation coefficients were used to calculate mean correlation matrices.”

Comment 7 - In the discussion, it could be useful to relate IC1 and IC2 to well-established neuroanatomical/molecular knowledge of the serotoninergic system. Did the authors expect the IC1 and IC2 anatomical distributions? is there a plausible biological reason as to why the time courses of BPnd variations would be somehow different between IC1 and IC2?

We appreciate the reviewer’s insightful comment and agree on the importance of relating IC1 and IC2 to well-established neuroanatomical and molecular knowledge of the serotonergic system.

In our discussion, we noted that IC1 primarily encompasses subcortical structures such as the brainstem, midbrain, and thalamus. These regions are consistent with areas housing dense serotonergic projections originating from the raphe nuclei, the primary source of serotonin release. In contrast, IC2 involves limbic and cortical regions - including the striatum, amygdala, cingulate, insular, and prefrontal cortices - which are key targets of the serotonergic pathways. This anatomical distinction aligns with the hierarchical organization of the serotonergic system, where the brainstem nuclei exert both local and distal serotonergic modulation.

The observed differences in the temporal dynamics of the binding potential (BP_ND_) variations between IC1 and IC2 likely reflect the distinct functional roles of these regions within the serotonergic network. The more immediate changes in IC1 could be attributed to the direct effect of MDMA on the raphe nuclei, leading to rapid serotonin release in subcortical structures. In contrast, the delayed changes in IC2 may reflect downstream modulation in cortical and limbic regions involved in processing more complex emotional and cognitive functions.

That said, while these interpretations are plausible based on current neuroanatomical and functional knowledge, the exact biological mechanisms underlying the differential time courses remain unclear. As discussed in the manuscript, future studies incorporating direct, simultaneous measurements of serotonin levels and imaging data will be essential to fully elucidate the temporal and spatial dynamics of serotonin transmission in these regions. We have revised to better highlight this limitation in the discussion section (page 17) as an important area for further investigation:

“Our results demonstrate that compared with FC, MDMA induces more pronounced changes in MC, particularly in regions associated with the SERT subcortical network. The distinct temporal dynamics of BPnd variations between these components may reflect the hierarchical organization of the serotonergic system. Specifically, the raphe nuclei, as the primary source of serotonin, are likely to exert more immediate modulation on posterior subcortical structures (IC2), whereas downstream effects on limbic and cortical regions (IC1) may occur more gradually. While these findings align with current neuroanatomical and molecular knowledge, the precise biological mechanisms driving these temporal differences remain unclear. Future investigations are warranted to elucidate these mechanisms. Future studies combining direct measurements of serotonin levels with neuroimaging data will be critical to fully understanding these components’ distinct roles and temporal profiles in regulating serotonergic function.”

Comment 8 - In the discussion (physiological basis), could the authors detail the expected "time scale" in changes in SERT expression? How quickly can SERT expression change, especially under resting-state conditions? Is it reasonable to consider tracer fluctuations under rest conditions as biologically meaningful?

SERT regulation can occur over different time scales depending on the mechanism involved [7].

Acute, rapid changes (milliseconds to seconds): Protein-protein interactions with key regulatory proteins (e.g., syntaxin1A, neuronal nitric oxide synthase) can lead to rapid modulation of SERT surface expression [8-11]. These interactions often involve changes in transporter trafficking or conformational states and can occur within milliseconds to seconds. For example, syntaxin1A directly interacts with the N-terminus of SERT, influencing its availability on the plasma membrane within short timescales.

Intermediate time scales (seconds to minutes): Posttranslational modifications, such as phosphorylation by kinases (e.g., protein kinase C) or dephosphorylation by phosphatases, are known to influence SERT function and surface expression [12-14]. These processes are typically initiated in response to cellular signaling and occur over seconds to minutes, affecting the SERT trafficking dynamics and serotonin uptake capacity [15, 16].

Longer-term changes (minutes to hours): Longer-term regulation involves processes like endocytosis, recycling, or degradation of SERT. These pathways typically take minutes to hours and are often part of more sustained cellular responses to changes in neuronal activity or serotonin levels. Such changes are slower but contribute to the overall cellular homeostasis of SERT under prolonged stimulation.

Under resting-state conditions, where neurons are not subjected to rapid or dramatic fluctuations in neurotransmitter release or signaling, SERT expression and activity are generally stable but still subject to subtle fluctuations due to ongoing basal regulatory processes**.** Basal phosphorylation or low-level protein-protein interactions can still dynamically modulate SERT trafficking and function, albeit at a lower intensity than under stimulated conditions. These fluctuations, although smaller in magnitude, may reflect fine-tuning of serotonin homeostasis and can occur on shorter timescales (seconds to minutes).

Biological Relevance of Tracer Fluctuations at Rest:

It is reasonable to consider that tracer fluctuations under resting conditions could reflect biologically meaningful variations in SERT expression and function. Even subtle shifts in SERT surface availability or activity can impact serotonin clearance and signaling, given the fine balance required to maintain serotonergic tone. These fluctuations may reflect intrinsic neuronal variability or ongoing homeostatic adjustments to maintain optimal neurotransmitter levels or serve as early indicators of adaptive responses to environmental or physiological changes before more overt modifications in transporter expression or activity become apparent.

In summary, while SERT expression can change rapidly in response to signaling events (milliseconds to minutes), even under resting-state conditions, subtle regulatory fluctuations can be biologically meaningful. These fluctuations likely reflect ongoing regulatory adjustments essential for maintaining serotonergic balance and should not be disregarded as noise, particularly in experimental measurements using tracers.

We added the following paragraph to the discussion (page 16):

**“**In addition, SERT regulation occurs over multiple time scales, ranging from milliseconds to hours, depending on the mechanism involved [31]. Rapid changes in SERT surface expression can be mediated by protein-protein interactions or posttranslational modifications [32, 33], such as phosphorylation, which occur on a timescale of milliseconds to minutes. These processes dynamically modulate surface availability and function, allowing fine-tuned regulation of serotonin uptake even under resting-state conditions. Additionally, while slower processes involving endocytosis, recycling, and degradation typically occur over minutes to hours, subtle fluctuations in SERT trafficking and activity can still occur under basal conditions. These minor yet biologically relevant changes likely reflect ongoing homeostatic regulation essential for maintaining serotonergic balance. Therefore, tracer fluctuations observed during resting-state measurements should not be dismissed, as they may represent meaningful variations in SERT regulation that contribute to the fine control of serotonin clearance.**”**

Comment 9 - In the discussion, the SERT network results should be commented on more extensively, as there is now only a generic reference to MC changes being stronger than FC ones, without spatial reference to the SERT network (while only negative salience network results are referenced explicitly instead, making the paragraph a bit confusing).

We expanded the discussion to accommodate a more thorough contemplation of this network. This revised paragraph (page 17) directly addresses the spatial aspects of the SERT network, highlighting the specific regions involved in serotonergic connectivity and contrasting molecular and functional connectivity changes induced by MDMA.

Comment 10 - Figure 3; I'd switch left and right charts in the bottom panel (last row only), to keep the SERT network always on the left of the Figure.

We agree with the suggestion and changed the figure accordingly.

Comment 11 - Figure 4: I'd add FC decreases to the figure, to allow the reader to compare BPnd, MC, and FC changes more easily and I'd add a horizontal line at the equivalent of e.g. Z-1.96 (or similar) so that it is clear which measures/regions display significant changes.

We prefer to keep the figure focusing on the two analyses of PET alterations, since we want to emphasize their complementarity in the context of PET specifically. However, we added lines indicating significances, in line with the reviewer’s suggestion.

Comment 12 - In Figure 5D, the y-axis mentioned FC but I suppose it should mention MC.

We amended the figure accordingly, together with the changes to the names of the networks implemented across the manuscript.

(1) Marciano, S., et al., Combining CRISPR-Cas9 and brain imaging to study the link from genes to molecules to networks. Proc Natl Acad Sci U S A, 2022. 119(40): p. e2122552119.

(2) Ionescu, T.M., et al., Striatal and prefrontal D2R and SERT distributions contrastingly correlate with default-mode connectivity. Neuroimage, 2021. 243: p. 118501.

(3) Ionescu, T.M., et al., Neurovascular Uncoupling: Multimodal Imaging Delineates the Acute Effects of 3,4-Methylenedioxymethamphetamine. J Nucl Med, 2023. 64(3): p. 466-471.

(4) Ionescu, T.M., et al., Elucidating the complementarity of resting-state networks derived from dynamic [(18)F]FDG and hemodynamic fluctuations using simultaneous small-animal PET/MRI. Neuroimage, 2021. 236: p. 118045.

(5) Walker, M., et al., In Vivo Evaluation of 11C-DASB for Quantitative SERT Imaging in Rats and Mice. J Nucl Med, 2016. 57(1): p. 115-21.

(6) Walker, M., et al., Imaging SERT Availability in a Rat Model of L-DOPA-Induced Dyskinesia. Mol Imaging Biol, 2020. 22(3): p. 634-642.

(7) Lau, T. and P. Schloss, Differential regulation of serotonin transporter cell surface expression. Wiley Interdisciplinary Reviews: Membrane Transport and Signaling, 2012. 1(3): p. 259-268.

(8) Haase, J., et al., Regulation of the serotonin transporter by interacting proteins. Biochem Soc Trans, 2001. 29(Pt 6): p. 722-8.

(9) Quick, M.W., Regulating the conducting states of a mammalian serotonin transporter. Neuron, 2003. 40(3): p. 537-49.

(10) Ciccone, M.A., et al., Calcium/calmodulin-dependent kinase II regulates the interaction between the serotonin transporter and syntaxin 1A. Neuropharmacology, 2008. 55(5): p. 763-70.

(11) Chanrion, B., et al., Physical interaction between the serotonin transporter and neuronal nitric oxide synthase underlies reciprocal modulation of their activity. Proc Natl Acad Sci U S A, 2007. 104(19): p. 8119-24.

(12) Qian, Y., et al., Protein kinase C activation regulates human serotonin transporters in HEK-293 cells via altered cell surface expression. J Neurosci, 1997. 17(1): p. 45-57.

(13) Ramamoorthy, S., et al., Phosphorylation and regulation of antidepressant-sensitive serotonin transporters. J Biol Chem, 1998. 273(4): p. 2458-66.

(14) Jayanthi, L.D., et al., Evidence for biphasic effects of protein kinase C on serotonin transporter function, endocytosis, and phosphorylation. Mol Pharmacol, 2005. 67(6): p. 2077-87.

(15) Steiner, J.A., A.M. Carneiro, and R.D. Blakely, Going with the flow: trafficking-dependent and -independent regulation of serotonin transport. Traffic, 2008. 9(9): p. 1393-402.

(16) Lau, T., et al., Monitoring mouse serotonin transporter internalization in stem cell-derived serotonergic neurons by confocal laser scanning microscopy. Neurochem Int, 2009. 54(3-4): p. 271-6.